# Effect of UHPH and Sulphur Dioxide Content on Verdejo Vinification: Sensory, Chemical, and Microbiological Approach After Accelerated Aging Test

**DOI:** 10.3390/microorganisms13112623

**Published:** 2025-11-19

**Authors:** Miquel Puxeu, Carlos Sánchez-Mateos, Inés Horcajo-Abal, Mercè Sunyer-Figueres, Victoria Castillo, Daniel Fernández-Vázquez, Alejandro Suárez, Natalia Santamaría, Enric Nart, Sergi de Lamo, Antonio Morata, Immaculada Andorrà

**Affiliations:** 1Parc Tecnològic del Vi, Carretera de Porrera Km. 1, 43730 Falset, Spainimma.andorra@vitec.wine (I.A.); 2enotecUPM, Chemistry and Food Technology Department, ETSIAAB, Universidad Politécnica de Madrid, Avenida Complutense S/N, 28040 Madrid, Spain

**Keywords:** Ultra-High-Pressure Homogenization (UHPH), sterilization, white must, wine, sulphur dioxide, microbiological analysis, chemical analysis, accelerated aging

## Abstract

Ultra-High-Pressure Homogenization (UHPH) is increasingly accepted by alimentation industries as a methodology to prevent microbial contamination with minimal impact on food organoleptic characteristics. Since 2022 the International Organization of Vine and Wine allows the use of high pressure (over 200 MPa), applied continuously, in winemaking. While previous works reported the impact of UHPH technology on must microbiology and color; in the present work, the effect of UHPH in Verdejo is investigated, evaluated, and compared with the use of different amounts of sulphur dioxide from a sensorial, chemical, and microbiological point of view. Our findings indicate that combining low doses of sulphur dioxide with UHPH improves wine quality, increasing the floral and overall assessment and decreasing the amount of aging flavors after an accelerated aging test. This study provides new insights into the effect of UHPH on wine quality and, specifically, on how the aging potential contributes to a better understanding of the impact of UHPH technology on the shelf life of wines.

## 1. Introduction

Sulphur dioxide (SO_2_) is the most common antioxidant and antibacterial agent used by the wine industry. Winemakers worldwide and food processing industries, in general, use it to prevent wine from oxidation, ensure stability from a microbiological point of view, and protect must and wine during all the processes of polyphenol aromatic damage and spoilage. However, some studies point to SO_2_ as potential cause of adverse health effects like abdominal pain, urticaria, headaches, and toxicity in vital organs [1]. In the last two decades, wine producers have been studying how to produce high-quality wines using less sulphur dioxide or even SO_2_-free processes, according to the preferences of consumers who are looking for healthy products [2]. Additionally, the International Organization of Vine and Wine (OIV) has gradually decreased the maximum limit of the total SO_2_ content in wines, which is currently at 150 mg/L for red wines and 200 mg/L for white wines, with some exceptions depending on the sugar content (OIV Resolution OENO 9/98).

In the last decade, most studies have focused on decreasing SO_2_ levels in wines through chemical methods, using alternative substances or agents in different steps of the winemaking process which could mimic SO_2_ activity. Tannins or yeast that produce and release glutathione into grape must or wine are examples of alternative antioxidants used by wineries that desire to minimize the use of SO_2_ for winemaking [3,4]. In fact, the oxidative protection and antimicrobial effect of adding tannins to wine are well documented [5]. Additionally, in terms of microbiological stability, including the control of wild microorganism populations, some agents like dimethyldicarbonate, lysozyme, or phenolic compounds have been tested with satisfactory results [6]. From an economic and technical point of view, the above-mentioned chemical techniques used to ensure wine microbiological stability and to prevent oxidation were considered more versatile than the use of physical methods [1,7]. On the other hand, the main advantage of using physical methods to improve microbiological stability is the absence of products added to the wine that could interfere with their organoleptic properties [7]. Some of the existing physical techniques which could replace the use of SO_2_ during the winemaking process or decrease its use and that have a direct effect on microorganism presence are pulsed electric fields (PEFs), ultraviolet (UV) radiation, high hydrostatic pressure (HHP), and flash-pasteurization [8,9,10]. Implementing new technologies in the winemaking process is a natural consequence of recent advances and research findings [11,12]. Recently, Ultra-High-Pressure Homogenization (UHPH) has emerged as a new physical technique to reduce or eliminate wild microorganisms in must or wine, ensuring microbial stability and the inactivation of oxidative enzymes like polyphenol oxidase (PPO), contributing to producing wines with improved sensory quality and longer shelf life while decreasing the need for sulphur dioxide [13,14,15,16].

UHPH is the natural evolution of the high-pressure process (HPP) with the advantage over the latter of being a continuous process. HPP uses pressures of 200 MPa or less while UHPH involves pressures higher than 200 MPa [17,18]. UHPH technology was accepted by the OIV in 2020 by the OIV-OENO resolution 594B-2020. What makes this technology interesting to winemakers is that it does not involve exposing the wine to thermal conditions known to affect wine’s organoleptic properties. Therefore, winemakers may continue producing high-quality wines with less SO_2_ while avoiding the negative consequences of alternative physical or thermal treatments [13]. High-pressure technologies have been used in other food manufacturing industries, such as the production of milk or fruit juices, since 1990 [18,19].

Oxygen (O_2_) concentration plays a fundamental role in wine quality outcomes, since the dissolved oxygen can have both good and bad consequences depending on timing in the winemaking process and existing concentrations [20]. For instance, the addition of oxygen during fermentation has a positive effect. In fact, oxygen deficiencies are among the main causes of fermentation interruptions, and the addition of at least 10–15 mg/L is often necessary [21]. However, at high O_2_ concentrations, and depending on other factors such as pH, the concentration of certain phenolic compounds, the presence of exogenous antioxidants such as SO_2_ or ascorbic acid, and storage temperature, wine can become oxidized [22].

The presence of certain chemical compounds such as methional, eugenol, sotolon, and 2,3,4-trimethyl-1,3-dioxane has been associated with oxidative degradation [23]. These compounds are associated with aromas described as “honey”, “dried fruit”, “farm feed”, “woody-like”, “cooked”, “oxidized apple”, and “aldehyde”; some of these descriptors have a negative impact on the aromatic composition of wine [22]. While the addition of sulphur dioxide is currently the mainstream technique used to prevent wine oxidation, the UHPH technique presents a promising, less-invasive alternative to combat the aging process of wine which is potentially more effective because of the direct and intense inactivation of polyphenol oxidases (PPOs) [24].

This study evaluates the effectivity of UHPH technology combined with ranging SO_2_ concentrations to prevent oxidation processes and microbial spoilage in winemaking. To evaluate the shelf life of wines produced under different combinations of UHPH and SO_2_, the chemical and microbiological properties of grape must as well as the sensory, chemical, and microbiological properties of wine were evaluated after an accelerated aging process.

## 2. Materials and Methods

### 2.1. Must Obtention and Preparation

The grapes were hand-picked from *Vitis vinifera* sp. verdejo vines in a commercial vineyard belonging to Bodegas Beronia (Rueda, Spain) during the 2024 harvest period. The fruit was harvested directly into 15 kg boxes and transported under refrigeration (at 4 °C) to the VITEC experimental winery in Tarragona (Spain). Once at the winery, the must was obtained using a Mini Ullrich Willmes Weintechnik press (Lampertheim, Germany) with a 300 kg whole-grape capacity and immediately transferred into 50 L steel tanks. The cold settling was carried out at 14 °C for 24 h using a Lafazyme CR enzyme at 1 g/hL (Laffort^®^, Floirac, France). A basic chemical characterization of the must was performed after clarification, and then the must was divided into two different sets. The first set underwent an UHPH technology process and was then divided into six independent tanks of 30 L, corresponding to 3 SO_2_ doses, with 2 repetitions each: UHPH SO_2_-free (×2), UHPH 50% SO_2_ (×2), and UHPH SO_2_ (×2). The second set did not undergo UHPH and was also divided into six independent tanks of 30 L named SO_2_-free (×2), 50% SO_2_ (×2), and SO_2_ (×2). The nomenclature “free” indicates that no sulphur dioxide was added at the grape, must, or wine stages, the nomenclature “50% SO_2_” indicates that the grapes and must were treated with 3 g/hL of sodium metabisulphite, and “SO_2_” indicates that the grapes and must were treated with 6 g/hL of sodium metabisulphite. A total number of six treatments and twelve tanks were studied.

### 2.2. Chemical Analysis

The chemical characterization of the must and wines was performed as follows. For must samples, the density, brix, potential alcohol strength, total acidity (expressed as tartaric acid), and pH, as well as the ammonia, primary amino nitrogen, and yeast-assimilable nitrogen contents were determined by infrared spectroscopy (FTIR) using a WineScan^TM^ instrument (FOSS, Hilleroed, Denmark), and the glucose–fructose and malic acid contents were determined via enzymatic reaction using a Y200 instrument (Biosystems S.A., Barcelona, Spain). For wines, the alcoholic degree, total acidity (expressed as tartaric acid), volatile acidity (expressed as acetic acid), pH, and free and total sulphur dioxide contents were determined using infrared spectroscopy (FTIR), WineScan^TM^. The organic acid content (malic and lactic acids) was determined using the Y200 Biosystems equipment for enzymatic reactions (Biosystems S.A., Barcelona, Spain). For the colorimetric measurements, 10 mm path-length plastic cells were used to measure absorbances at 420 nm for yellow color contribution and at 450 nm, 520 nm, 570 nm, and 630 nm for the Cie Lab coordinates using a Helios-α spectrophotometer (Thermo Fisher Scientific, Waltham, MA, USA). Turbidity was measured with a Hach TL2310 turbidimeter (Hach, Loveland, CO, USA). Total phenols were determined using the Folin–Ciocalteu assay method (OIV-MA-AS2-10) with some modifications. Briefly, 100 μL of sample, 500 μL of Folin–Ciocalteu reagent, and 2 mL of a sodium carbonate solution (1.88 M) were mixed with water to obtain a final volume of 10 mL. The solution was stocked for 30 min for the reaction to take place and stabilize. Finally, the absorbance was measured at 750 nm by a Helios-α spectrophotometer (Thermo Fisher Scientific, Waltham, MA, USA). All analyses were performed according to the methods recommended by the Compendium of International Methods of Analysis—Organization of Vine and Wine (OIV) 2020 [25].

### 2.3. UHPH Treatment Conditions

Clarified must was processed using a UHPH system built under the patent by Universitat Autònoma de Barcelona (EP2409583B1) and capable of continuous UHPH treatment at 180 L/h (Ypsicon Advanced Technologies, Barcelona, Spain). The equipment used a tungsten carbide valve and was deployed in the VITEC experimental winery. The processing parameters were as follows: flow rate—180 L/h at 200 ± 10 MPa, inlet temperature—15 °C, in-valve temperature—65–78 °C for only 0.02 s, and outlet temperature—15 °C.

### 2.4. Microbiological Analysis

Common oenological microorganisms (yeast, lactic and acetic bacteria, and *Brettanomyces*) were quantified by plate count and Real-Time Polymerase Chain Reaction (RT-PCR). For the plate count, wine was inoculated into different solid media depending on the microorganism and incubated for different times and temperatures according to the methods described in the OIV (2020) [25], as follows: for the yeast count, YPDA medium (2% glucose, 2% peptone, 1% yeast extract, and 2% agar, *w*/*v*; Panreac, Barcelona, Spain) was used, incubated at 25 °C for 4 days; for the acetic acid bacteria count, GYC medium (5% glucose, 3% calcium carbonate, 1% yeast extract, 2.5% agar, 0.01% natamicyn, and 0.0013% penicillin, *w*/*v*; Panreac, Barcelona, Spain) was used, incubated at 25 °C for 6 days; for the lactic acid bacteria count, MRSA media (Sharlau, Barcelona, Spain) was used, incubated at 30 °C for 10 days, and for the *Brettanomyces* count, *Brettanomyces* Agar (Sharlau, Barcelona, Spain) was used, incubated at 25 °C for 7 days.

For the RT-PCR, 50 mL of wine or must was centrifuged to collect the cells, which were then washed, and DNA was isolated using the DNeasy Plant MiniKit (Qiagen, Valencia, CA, USA) with some modifications to improve cellular lysis. Prior to extraction, the cells were suspended, added to a 2 mL screw-cap tube with 1 g of 0.5 mm-diameter glass beads, and lysed using an MBB-16 Mini-Beadbeater (BioSpec Products, Cambridge, UK) (5 cycles of 30 s in the Mini-Beadbeater + 30 s on ice) [26]. RT-PCR was performed using a mix of the cells with 2 µL of DNA, 0.4 µL of each primer, 0.08 µL of ROX (SYBR Premix Ex II (TaKaRa Bio Inc., Shiga, Japan)), 10 µL of SYBR Green, and 2.12 µL of sterile Milli-Q water. The RT-PCR reactions were performed with different primers, according to each microorganism, to quantify the total yeast content, YEASTF and YEASTR [26]; lactic bacteria content, WLAB1 and WLAB 2 [27]; acetic acid bacteria content, AQ1F and AQ2R, and *Brettanomyces* content, DBRUX F and DBRUX R [28]. Amplification was conducted using a QuantStudio3 Real-Time PCR system (Thermo Fisher Scientific), following the amplification program, as follows: 50 °C for 2 min, 95 °C for 10 min, 40 cycles of 95 °C for 15 s, 60 °C for 1 min, and 72 °C, 30 s. Samples were analyzed in triplicate, and for all reactions, two negative controls were used (water as a non-template control (NTC) and an extraction negative control) and one sample of DNA was used as the positive control.

### 2.5. Particle Size After UHPH by Laser Diffraction

Particle size measurements were performed by laser diffraction, using the Malvern Mastersizer 2000^®^ (Malvern Instruments Ltd., Malvern, UK). Samples were previously diluted with distilled water until the appropriate laser obscuration values were obtained (5–10%). The refractive indexes for the sample and water were set at 1.340 and 1.333, respectively. The particle size distribution was characterized by the D50 and D90 (particle diameter at 50 and 90% in the cumulative distribution) and d3.2 (surface area average diameter) and d4.3 (volume moment mean) parameters. Determination was performed on the third day after UHPH treatment. Measurements were performed in triplicate.

### 2.6. Particle Size and Colloidal Structure by Atomic Force Microscopy (AFM)

Air-dried must samples of 10 µL were scanned using a Nano-Observer AFM (Concept Scientific Instruments, Les Ulis, France) working in resonant mode to obtain the topography and sizes of colloidal particles included in the juice. The silicon cantilever used to analyze the surface had a strength of 1 N/m and a 15 nm nominal diameter (model Fort, AppNano, Mountain View, CA, USA). The frames used ranged from 100 to 10 µm and the scanning speed ranged from 2 to 0.5 L/second. For the UHPH treatments, more than 30 colloidal particles were measured.

### 2.7. Microfermentation Process

After the application of different concentrations of sulphur dioxide or UHPH, musts were fermented in a recirculating bath of cool water at 17 °C to avoid drastic changes in the fermentation temperature. Fifty liters of white must were settled at 12 °C in a temperature-controlled room and favored with the addition of 1.2 g/hL of LAFAZYM^®^ CL (Agrovin S.L., Alcázar de San Juan, Spain) pectolytic enzymes. Thirty liters of clarified must was initiated with 20 g/hL Fermivin^®^ PDM (Oenobrands SAS, Montferrier-sur-Lez, France) yeast, applying nutrition with Actimax Varietal (Agrovin S.L., Spain) at a density of 1.070 g/cm^3^ and with Actimax Plus (Agrovin S.L., Alcázar de San Juan, Spain) when the density reached 1.030 g/cm^3^. Then, the must was monitored by classical analytical techniques (density and temperature). Once residual sugar concentrations were lower than 0.5 g/L, alcoholic fermentation was considered completed and wines were stored under controlled temperatures. Before bottling, the obtained wines were clarified with the addition of 40 g/hL bentonite Microcol^®^ (Laffort^®^, Floirac, France), cold-stabilized at 4 °C for 5 days and filtered through 0.80 µm pore sizes formed by a pleated polypropylene cartridge filter by Parker Hannifin (Cleveland, OH, USA). Experimental cellar facilities were cleaned and disinfected by hot water at 80 °C for 30 min and Ox-virin for all the surfaces (Grupo OX, Cuarte, Spain).

### 2.8. Accelerated Aging Process

Forced oxidation processes were performed following the conditions and protocols described by some authors [29,30]. Wine bottles with a 0.75 L capacity were separately oxidized by forcing an air stream against the wine under controlled laboratory conditions until oxygen saturation, 8.50 ppm [30]. The concentrations of dissolved oxygen for all samples submitted at accelerated aging were measured with a NomaSense P300 instrument (Vinventions France SAS, Ribesaltes, France) until the wine reached 100% saturation. Each saturated wine was kept in a closed glass bottle and incubated at 60 °C for 10 days. This forced aging implementation has been used previously in similar experiments with white wines [30,31]. After oxidation, a basic chemical characterization, fermentative aroma and oxidation aroma compounds by were conducted by GC-MS.

### 2.9. Analysis of Fermentation and Oxidation Aroma Compounds by Gas Chromatography with Mass Spectroscopy (GC-MS)

Fermentative aromas, oxidation resistance, and adequate aging potential are ongoing challenges in white wine production. Fermentative volatile compounds were analyzed as described by Torrens et al. [32]. An Agilent Technologies 7890A gas chromatograph (Palo Alto, CA, USA) was used, coupled to a triple quadrupole mass detector 5975 MSD equipped with a Headspace-HS injection MultiPurpose Autosampler for GC-MSD (Gerstel, Eberhard-Gerstel-Platz, Germany). The DB-WAX-UI column and pressure of (60 × 250 μm × 0.25 μm) were used at a flow rate of 1.6 mL/min and a pressure of 25 psi. Helium of purity greater than 99% was used as a carrier gas. The injector temperature was 250 °C for 1 min and then 235 °C to 260 °C/min. The oven temperature started at 40 °C for 1 min, rising to 225 °C at 15 °C/min and ending at 260 °C at 100 °C/min. The mass spectrometer was operated in electron ionization mode at 70 eV. Data acquisition and analysis was performed using Agilent Technologies MSD Chemstation software(version F.01.032357), performing a full-scan analysis (*m*/*z* 50–350). The volatile compounds were identified by comparison of their mass spectra, with the help of the NIST library. The quantification was carried out by the internal standard method (IS). The individual volatile compounds were quantified with the factor of response of 2-octanol (100 µL 2-octanol at a 1.000 ppm concentration in 10 mL of wine sample) and expressed as the equivalent of 2-octanol in μg/L. All analyses were carried out in duplicate.

Evolution compounds were determined using an Agilent Technologies 7890A gas chromatograph coupled to a triple quadrupole mass detector (MS) equipped with a liquid injection—LI MultiPurpose Autosampler for GC—GC/MS (Gerstel, Eberhard-Gerstel-Platz, Germany) according to [33]. The VF-200 ms (30 m × 250 μm × 0.25 μm) Agilent J&W GC Columns were used at a constant flow rate of 1.7 mL/min. Helium of purity greater than 99% was used as a carrier gas. A total of 5 µL was injected in Splitless mode with a pressure of 16 psi, a septum purge flow rate of 3 mL/min, and a Splitless time of 1 min. The injector temperature was 180 °C for 1 min and then 260 °C to 250 °C/min. Oven temperature started at 40 °C for 1 min, rising to 220 °C at 10 °C/min and ending at 270 °C at 100 °C/min. The mass spectrometer was operated in electron ionization mode at 70 eV. Data acquisition and analysis were performed using the Agilent Technologies MSD Chemstation software, performing a full-scan analysis (*m*/*z* 50–350) comparing with each of the reference compounds, which were calibrated against external (r^2^ > 0.99) and internal (20 µL 2-methylpentanal) standards. All compounds were from Fluka (Sigma-Aldrich, Buchs, Switzerland). The oxidative compounds were separated by derivatization with O-(2,3,4,5,5,6-pentafluorobenzyl) hydroxylamine (1 mL of 10 mg/mL) to obtain the derivatized oximes of the alkenals, aldehydes, ketones, lactones, and furans. These were then extracted with Bond Elut ENV cartridges (Agilent, Palo Alto, CA, USA) (200 mg) and collected with dichloromethane. Subsequently, the samples were evaporated from the solvent to 250 µL of concentrated extract to be injected and analyzed by GC-MS. In this study, the wine’s ability to resist oxidation was evaluated by comparing different antioxidant strategies, including the use of sulphites and ultra-high-pressure homogenization (UHPH).

### 2.10. Sensory Analysis

All the wines samples, produced according to the conditions described above, were sensory-analyzed in a normalized ISO 8589:2007 [34] room. The judges of the tasting panel were selected, trained, and qualified following the normative ISO 8586:2023 [35]. The training consisted in familiarization with sensory evaluation procedures, such as recognizing the basic tastes, aromas, and textures, as well as using standardized scales and rating systems to describe these attributes and their intensities. Once the initial training is complete, the active panel judges are regularly monitored to ensure their skills remain sharp and their tasting ability is consistent over time. Finally, a quantitative descriptive analysis (QDA) was carried out by 8 judges (age range: from 26 to 45 years old, 2 women and 6 men). The tasting panel used in this work has been continuously tasting since 2021 at a frequency of 2 tastings per week. Informed consent for participation was obtained from all subjects involved in the study, and no informed consent statement was required.

A test to determine if there were any significant differences between treatments was performed as follows. The two samples from different treatments were presented to the panel blindly (labeled with 3-digit random codes) using William’s design for the sample sets assigned to each judge. The sensory attributes were divided into 3 groups: color (intensity and evolution), aroma (intensity and profile), flavor (sourness, astringency, unctuosity, bitterness, persistence, dryness, tannic intensity, and burning), and overall assessment punctuation was also considered. Among the aroma profiles, fruity aromas (white, tropical, and citrus), spicy, balsamic, floral, chemical, and vegetal were considered. The attributes were rated on a linear scale ranging from 1 (absence) to 5 (maximum intensity). From the list of attributes, the judges only rated those that they perceived to be within the attribute’s scale. The wine ratings were collected using Compusense^®^ Cloud software (Compusense Inc., Guelph, Ontario, Canada-Version 25.0.16).

### 2.11. Statistical Analysis

Means and standard deviations were calculated, and significant differences were determined by one-way analysis of variance (ANOVA) by the Tukey (HSD) test using the XLSTAT package (2016.01) for Excel software (Version 2510). Significance was set at *p* < 0.05.

## 3. Results and Discussion

### 3.1. Effect of UHPH on Colloidal Particle Size and Must Structure Using AFM

The dried juice of Verdejo grapes were scanned after UHPH treatment during the 2023 (preliminary works) and 2024 (present study) harvest periods, and the average size of the particles was around 500 nm in the submicron range (Table 1, Figure 1 and Figure 2b), in agreement with previous works [24]. In the untreated must samples, a relatively homogeneous surface was documented, characterized by the presence of polyhedral fragments, indicating that no super-micron particles (max. size 843–973 nm, Table 1) were present after the UHPH treatment, which is evidence that all grape microorganisms such as yeasts and bacteria were destroyed by the treatment. In comparison, the initial colloidal size of the juice particles was 2 µm on average (range 1.1–3.7 µm, Table 1; Figure 2a). After the UHPH processing, a regular colloidal structure can be observed by AFM with a uniform dispersion of colloidal particles at submicron scale (Figure 2b), which is also in agreement with previous research on grape juice [24,36]. The treated must exhibits a notable reduction in the quantity of these structures (Figure 1 and Figure 2b) compared to the control must (Figure 2a), suggesting a direct effect of the treatment on the composition of the must particles. These findings provide evidence that the UHPH treatment not only reduces the size of the fragments but also alters their distribution.

It is worth noting that, despite the existing fragments exceeding 200 nm in the UHPH-treated must, no adverse effects were observed in terms of nanosafety, given that the critical threshold is established below 100 nm [37,38]. This suggests that the UHPH treatment could be advantageous to produce musts with specific characteristics, without compromising the safety of the final product.

### 3.2. Chemical and Microbiological Composition of Must and UHPH Treatment

The results of the chemical characterization of grape juice after being clarified and submitted to the different treatments are shown in Table 2. No significant differences among treatments were found for degrees brix, glucose+fructose concentration, potential alcohol strength, total acidity, pH, ammonium and primary amino nitrogen, and L-malic acid. On the other hand, grape must color was significantly impacted by the UHPH technology and sulphur dioxide concentration. Higher absorbances at 420 nm were observed in samples not UHPH technology-treated and as the dose of sulphur dioxide was decreasing in all cases (Figure 3). The addition of sulphur dioxide as well as the dose had a direct impact on absorbance at 420 nm for all the studied treatments. In the treatments in which UHPH technology was applied, the color of grapes must had lower intensities compared with those without UHPH treatment, with the UHPH treatment group having a lower yellow and total coloration (Table 1 and Figure 4). The effect of sulphur dioxide on the color intensity of musts and wines has been reported before [39,40], and it was also observed in this study. In the case of the UHPH technology group, this decrease in the total coloration of musts is due to the antioxidant capacity of must juice for enzymatic oxidations by polyphenol oxidases (PPOs) [13]. Similar absorbances at 420 nm were obtained by UHPH-free SO_2_ and 50% SO_2_, reinforcing the idea that the present technology could help winemakers to reduce the use of sulphur dioxide without reducing the product quality and without a negative impact on organoleptic characteristics.

Microbiological concentrations of yeast, lactic acid bacteria, acetic acid bacteria, and *Brettanomyces bruxellensis* were determined for all the different treatments. For all the treatments where UHPH technology was applied, there was no detection of any of the oenological microorganisms—yeast, acetic and lactic acid bacteria, and *Brettanomyces bruxellensis*—by culture plate and qPCR. These results correspond to those obtained by other authors [24]. In the case of must without the application of UHPH technology, 10^5^ cells/mL for yeast, 10^2^ cells/mL for acetic acid bacteria, 10^2^ cells/mL for lactic bacteria populations were detected. *B. bruxellensis* was not detected. These results agree with previously published data [41]. Sulphur dioxide concentration has a direct effect on the initial populations, especially of bacteria, but without the possibility the UHPH technology has for obtaining sterilized must.

### 3.3. Fermentation Kinetics

Fermentation performance was assessed through daily measurements of density (Figure 4) and the residual sugar concentration at the end of fermentation (lower than 0.5 g/L). In all fermentations tested, the kinetics curves obtained showed typical shape without detecting stuck fermentations or many other alterations such as long early exponential phases produced by yeast acclimation and slow fermentation speed in the stationary phase due to the concentration of ethanol. Fermentations showed a short acclimatization period of 24 h, followed by an exponential phase that lasted several days under all treatments, after which the fermentation entered the stationary phase, completing the fermentation at around 15 days. Small differences were observed in the initial hours of the exponential phase, and the UHPH batches were the fastest ones regardless of suphur dioxide concentration. Despite this, these differences were not statistically significant. After thirteen days of alcoholic fermentation, the sugar concentration was below (0.5 g/L) in all the treatments, and the alcoholic fermentation was considered finished. Therefore, the results indicate that neither the UHPH treatment nor the SO_2_ concentration affected the must in a manner that influenced the fermentation performance of the inoculated yeast.

### 3.4. Chemical, Microbiological, and Sensory Composition of Wines After Alcoholic Fermentation

According to the basic parameters, no significant differences were found in parameters like alcohol, volatile acidity, total acidity, pH, and the L-malic and L-lactic acids of the studied wines coming from musts treated or not treated by UHPH with different SO_2_ concentrations (Table 3). On the other hand, significant differences were detected in color and oxidation-related parameters. Absorbance at 420 nm, often used as an indicator of oxidative browning (40), was significantly lower in the UHPH-treated samples with partial or SO_2_-free treatments (0.075–0.077) compared to the SO_2_ without UHPH treatment (0.131), suggesting a protective potential effect of UHPH against oxidative browning. The UHPH SO_2_-free sample showed lower oxidation (0.104) than the full SO_2_ treatment without UHPH technology, indicating that UHPH may reduce the need for SO_2_ under certain conditions. Absorbance at 420 nm was highest in the SO_2_-treated sample (0.131), followed by the 50% SO_2_ sample (0.105), while all UHPH treatments showed significantly lower intensities, especially the UHPH 50% SO_2_ sample (0.075), suggesting that UHPH may lead the prevention of polyphenol oxidation [13]. In terms of CIE Lab color parameters, all wines maintained very high Luminosity (L*) values (>97), reflecting the expected brightness of young wines. The b* coordinates (yellow–blue axis) were lower in the UHPH 50% SO_2_ and UHPH SO_2_ treatments than wine with SO_2_, indicating a lower oxidative effect and UHPH mitigation of browning reactions. UHPH treatment independent of SO_2_ concentrations had an important and significant impact on wine turbidity, with the UHPH SO_2_-free treatment showing low turbidity (1.0 NTU), while free or SO_2_-containing wines without UHPH showed higher turbidity (>7.0 NTU). These results suggest that UHPH may improve the clarity of wines. The Folin–Ciocalteu index, which is an estimate of the total phenolic content, was highest in the SO_2_-free and SO_2_ 50% treatments, 6.11 and 5.08, respectively, with samples obtained with UHPH treatment showing lower values, suggesting a partial degradation or structural modification of phenols [27,42].

Even though all musts treated by UHPH were sterilized to remove all oenological microorganisms, they can still be present after the alcoholic fermentation for several reasons. On one hand, yeasts inoculated during the alcoholic fermentation can be found in final wines. Moreover, microorganisms can be found on the equipment in the cellar (yeasts and lactic and acetic acid bacteria), that can contaminate the wine during the winemaking process. For this reason, the population conditions after alcoholic fermentation, clarification, and bottling were controlled. Details of all microbiological results after bottling are shown in Table 4. The results show that culturable populations of yeast, acetic acid bacteria, lactic acid bacteria, and *Brettanomyces bruxellensis* were below the detection limit (<1 CFU/mL) for all samples treated by UHPH (Table 4). In the case of samples untreated by UHPH, culturable yeast and acetic acid bacteria were detected in the 50% SO_2_ treatment. The populations of acetic acid bacteria, lactic acid bacteria, and *Brettanomyces bruxelensis* detected by RT-PCR were also lower than the detection limits (<20 cells/mL) for all the samples treated by UHPH. Only in the case of yeast, different populations (between 103 and 118 cells/mL) were found in all the samples. The RT-PCR technique detects the presence of DNA, which may come from either live or dead cells, so the population found can be either viable but not cultivable or dead. With regards to the samples not treated by UHPH technology and analyzed by RT-PCR, different populations of yeast were detected (from 109 to 22.825 cells/mL) and acetic acid bacteria at 56 cells/mL for 50% SO_2_ treatment. Wines produced from must treated with UHPH did not have microorganisms that came from the grape, but, like the non-treated wines, were susceptible to contamination during the winemaking process, as is shown in the case of 50% SO_2_ for yeasts. Therefore, the UHPH treatment reduces the sources of growth of non-desirable microorganisms, making wines more stable from a microbiological point of view.

The analysis of fermentative volatile compound groups after the alcoholic fermentation and oxidation conditions shows different behaviors depending on the studied treatment. Ultra-High-Pressure Homogenization (UHPH) and sulphur dioxide addition at different concentrations did not produce significant differences (*p* < 0.05) among samples in total ethyl esters, total acetates, and total fatty acids (Figure 5). This result suggests that, prior to oxidative stress, both UHPH and SO_2_ had a limited impact on the concentration of major aroma-active esters, alcohols, and fatty acids. However, a slight decrease in acetate esters was noted in the SO_2_-treated samples compared to the SO_2_-free samples, potentially reflecting early-stage ester hydrolysis or transformation [43]. These results indicate the neutral effect of UHPH treatment on the fermentative aromas produced by yeast during alcoholic fermentation. These results are in accordance with the results obtained by [44].

In terms of the oxidation markers of wines after alcoholic fermentation, the main aromatic compounds associated were analyzed and data considered the sum of all the compounds of each family: alkenals, strecker aldehydes, lactones, ketones, and aldehydes (Table 5). The alkenals family were formed by (E)-2-hexenal, (E)-2-heptenal, (E)-2-octenal, and (E)-2-nonenal, the strecker aldehydes family by 3-methylbutanal, 2-phenylacetaldehyde, and methional, the lactone family by γ- nonalactona and 3-methyl-2,4-nonadiona, furans by 3-hydroxy-4,5-dimethylfuran-2(5H)-one, and the aldehydes family by benzaldehyde.

Comparable values were obtained between the UHPH SO_2_-free and SO_2_ treatments, exhibiting comparable levels of oxidative aromas. These results indicate the potential of the UHPH technique to produce wines without SO_2_ with the same oxidation level as wines with the conventional use of SO_2_. The wine with a lower concentration of oxidative aromatic compounds was obtained with a conventional dose of SO_2_, and the wine with a higher oxidative aroma intensity was SO_2_-free, both without the use of UHPH.

The compounds that most strongly contribute to the high levels of oxidative aromas in wines with insufficient oxidation protection are aldehydes, particularly 2-phenylacetaldehyde and benzaldehyde, as shown in Table 5. These compounds are typically associated with notes of faded flowers, honey, and bitter almond. ANOVA results revealed significant differences in their concentrations among treatments: untreated or partially treated wines exhibited the highest levels, whereas SO_2_ and/or UHPH significantly reduced them. Overall, these aldehydes emerge as key contributors to the oxidative aroma profile, explaining the stronger sensory perception of oxidative notes in inadequately protected wines.

As shown in Figure 6a,b, the formation of oxidative aromas after fermentation was relatively low in wines treated only with SO_2_ (SO_2_), with values comparable to those obtained exclusively with UHPH (UHPH SO_2_-free). All other combinations displayed higher levels of initial oxidative aromas, with the SO_2_-free treatment showing the highest concentration. After the oxidation process, the SO_2_-free oxi. treatment showed the highest increase in oxidative aromas, followed by all other treatments without the use of UHPH (50% SO_2_ oxi. and SO_2_ oxi.). All treatments which combined the use of UHPH and sulphur dioxide (UHPH 50% SO_2_ oxi. and UHPH SO_2_ oxi.) showed lower concentrations of oxidative aromas.

A total of nineteen sensorial parameters were analyzed by the tasting panel. Between them, significant differences were found in two parameters, that being floral and overall assessment attributes (Figure 7). The best-rated wine by floral and overall assessment attributes was, in both cases, UHPH 50% SO_2_. As these results show, the combination of the UHPH technology with lower doses of sulphur dioxide could be a good option for wineries to produce wines with a lower concentration of sulphur dioxide without the loss of sensory characteristics and flavors. No significant differences were found for wines produced without the use of sulphur dioxide in floral and overall assessment attributes.

### 3.5. Effect of UHPH Technology Combined with SO_2_ Concentrations After Forced Aging

After the forced oxidation processes, analyses of the basic chemical characteristics, fermentative aromas, and oxidative aromas were carried out to evaluate the impact of the new technology combined with different doses of SO_2_ on the capacity of wines to fight against the accelerated oxidation process. According to the basic characterization, no significant differences were found between alcoholic degree, volatile acidity, total acidity, pH, and L-malic and L-lactic acid. The greatest impact of the accelerated oxidation on the basic parameters were in color characteristics, shown in Table 6 and Figure 8. Oxidation caused an increase in absorbance at 420 nm, resulting in a progressive darkening of the wine. This effect was particularly pronounced in untreated wines and those treated only with UHPH, which reached the highest values in all color bands. In contrast, wines treated with SO_2_, both individually and in combination with UHPH (even at reduced doses of 50%), showed significantly lower values, closer to those of the initial wine. These results confirm that SO_2_ is an effective protector against oxidative browning and that its combination with UHPH enhances this effect, allowing color stability to be maintained even with lower concentrations of sulphur dioxide.

Focusing only on the values after the forced oxidation process, the colors of the wines reveal significant differences related to oxidation depending on the treatment strategy used (UHPH, free, partial, and full use of SO_2_). Absorbance 420 nm, as an oxidative browning indicator, was highest in the UHPH SO_2_-free and SO_2_-free treatments (0.262 and 0.243, respectively), confirming their greater susceptibility to oxidation degradation. Simultaneously, samples treated with UHPH SO_2_ and SO_2_ without UHPH showed the lowest 420 nm values (0.131 and 0.111, respectively), reflecting higher oxidation resistance. These results align with the known antioxidant effect of SO_2_ and suggest that UHPH could be used as additional protection against oxidation processes once wines are bottled. Oxidative polymerization and browning were more pronounced in the absence of SO_2_. The L* values in the CIE Lab coordinates indicate a clear difference in brightness, with the samples of UHPH SO_2_ and SO_2_ showing higher values (>98), higher visual brightness, and less browning. On the other hand, UHPH SO_2_-free and SO_2_-free samples had significantly lower L* values, confirming their oxidation level due to the low concentration or absence of SO_2_. Regarding the b* coordinate, which indicates the color preservation, the highest values were, notably, shown for the UHPH SO_2_-free and SO_2_-free treatments (15.39 and 14.38, respectively), and the lowest values were found in the UHPH SO_2_ and SO_2_ treatments (8.82 and 7.15, respectively). No significant differences were found for the a* CIE Lab coordinate (red–green axis).

In contrast, the forced oxidation process reduced the levels of volatile ethyl esters and acetates across all treatments (Table 7). Notably, the UHPH SO_2_-free and SO_2_-free samples showed the largest decreases in both ethyl esters and acetates, suggesting an increased susceptibility to oxidative degradation without the presence of sulphur dioxide. In these treatments, the UHPH SO_2_-free treatment exhibited a dramatic drop in total ethyl esters (from 30.874 to 9.198 µg/L), highlighting the protective role of SO_2_ and the limited antioxidant capacity of UHPH without any other antioxidant agent preserving ethyl ester content under stress treatments like accelerating aging. Samples treated with SO_2_ or 50% SO_2_, in both the UHPH and conventional contexts, maintained significantly higher levels of ethyl esters during forced oxidation, with reductions between 25 and 75% of total ethyl esters compounds. These results indicated that partial SO_2_ addition, combined with UHPH, can offer a synergistic protective effect, being one of the most significant results of the present study. Total alcohol levels remained stable across treatments, with oxidative processes yielding no significant differences in alcohol concentrations between the SO_2_, 50% SO_2_, and SO_2_-free treatments, implying that alcohol compounds are less propitious to oxidative degradation than other compounds [32]. The percentage of total aroma loss for treatments without SO_2_ for the UHPH and no-UHPH treatments were 63% and 56%, respectively, 25% and 21% for the 50% SO_2_ UHPH and no-UHPH treatments, respectively, and finally, 35% and 28% for the UHPH SO_2_ and SO_2_ treatments. Overall, these findings suggest that while UHPH alone may not be enough for preventing the oxidative loss of key aroma compounds, its combination with reduced SO_2_ doses or the conventional use of SO_2_ could be a viable enological strategy to preserve wine aroma under oxidative stress, contributing to the development of more sustainable wine processing alternatives.

In relation to oxidative aromas, it was observed that they reached their highest values in wines without the application of suphur dioxide or UHPH, as we can see in Figure 6 and Table 5. In contrast, the combined treatment of UHPH and SO_2_ at 50% showed the greatest resistance to oxidation at the aromatic level, followed by the same treatment with SO_2_ at 100%. Overall, these results show that the application of UHPH in combination with SO_2_ is an effective strategy for delaying the appearance of oxidative notes, while also allowing for a reduction in the dose of sulfites without compromising the sensory stability of the wine.

As was expected, the accelerated oxidation process had an important effect on the concentration of oxidation markers for all the wines (Table 5). Figure 6b shows the oxidative aroma levels in the wines after forced oxidation. A marked increase was observed in the untreated wine (SO_2_-free oxid.), followed by the wine containing only 50% SO_2_ without UHPH. Intermediate values were found in wines treated with SO_2_ alone or UHPH alone. In contrast, the lowest concentrations were detected in wines treated with both techniques, as well as in those treated with 50% SO_2_ combined with UHPH. These results indicate that the combined application of SO_2_ and UHPH provides the greatest protection against oxidative aroma development, while SO_2_ and UHPH applied individually confer a similar, yet insufficient, level of protection to maintain a low oxidative profile. After oxidation, wines showed significantly higher concentrations of aldehydes (2-phenylacetaldehyde and 3-methylbutanal) and lactones, as confirmed by the ANOVA analyses. Among the aldehydes, the highest levels were observed in oxidized wines without SO_2_ and without UHPH treatment, whereas the application of SO_2_, UHPH, or their combination resulted in significantly lower concentrations. A similar trend was observed for γ-nonalactone: wines lacking protective treatments accumulated the highest amounts, while those treated with SO_2_ and/or UHPH exhibited reduced levels. These results demonstrate that both SO_2_ addition and UHPH processing mitigate the oxidative formation of aldehydes and lactones, thereby preventing the development of off-aromas such as dried fig, peach, or coconut notes and contributing to a more stable aromatic profile.

This study demonstrates that the combined application of SO_2_ (at both 100% and 50%) with UHPH was the most effective strategy, leading to the lowest concentrations of oxidative aroma compounds and providing greater resistance to their formation over time. When applied individually, SO_2_ or UHPH also reduced aldehyde and lactone accumulation, although their protective effect was less sustained compared to the combined treatment. In contrast, wines produced without any treatment consistently exhibited the highest levels of oxidative aromas. Finally, this study highlights the importance of combining strategies to replace or remove sulphur dioxide in winemaking to obtain high-quality wines. The effects of UHPH technology from a sensory and volatile perspective had not been studied yet in relation to their potential for higher stability and shelf life.

## 4. Conclusions

This study provides clear evidence of the potential of UHPH technology for the wine sector to produce healthy and good-quality wines without the need for using additives to increase the optimal period of consumption. The singularity of the present study lies in addressing all the efforts to evaluate the effect of a new technology on the chemical, sensory, and microbiological properties of bottled wines before and after being submitted to forced oxidation treatment. Results clearly show that UHPH technology combined with lower doses of sulphur dioxide could be a potential strategy to obtain sensorially and microbiologically stable wines with high resistance to unfavorable environmental conditions before their consumption. The impact of UHPH on preserving fermentative aromas is not significant but could help winemakers to obtain less colored grape musts, more stable wines from a microbiological point of view, and wines with higher resistance to enzymatic oxidation due to the inactivation of enzymes.

Future research will focus on studying other strategies like the use of oenological tannins or vegetal extracts to protect wines after the treatment of must by UHPH. Consumers around the world are looking for more sustainable, healthy wines with less additives used, and UHPH technology fits very well for this purpose. However, the initial research status and the high price of this technology are restrictions to its wide use in the wine sector. Furthermore, the potential of the UHPH technology exceeds that of other physical technologies for must or wine treatments, especially due to its non-thermal effect, which contributes to preserving the sensory properties of the product.

## Figures and Tables

**Figure 1 microorganisms-13-02623-f001:**
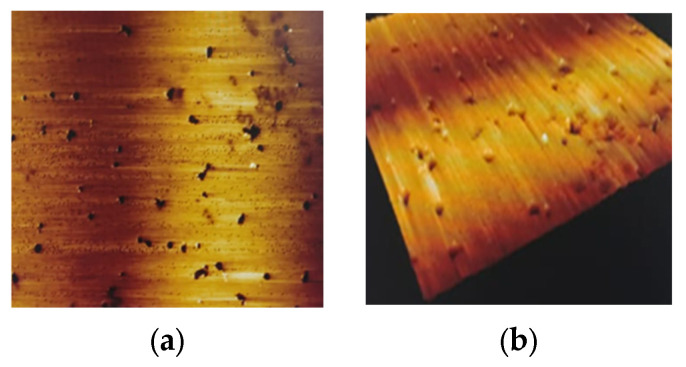
Atomic Force Microscopy (AFM) resonant scanning at 25 µm frame of Verdejo must treated with UHPH (**a**) 2D Topography; (**b**) 3D Topography.

**Figure 2 microorganisms-13-02623-f002:**
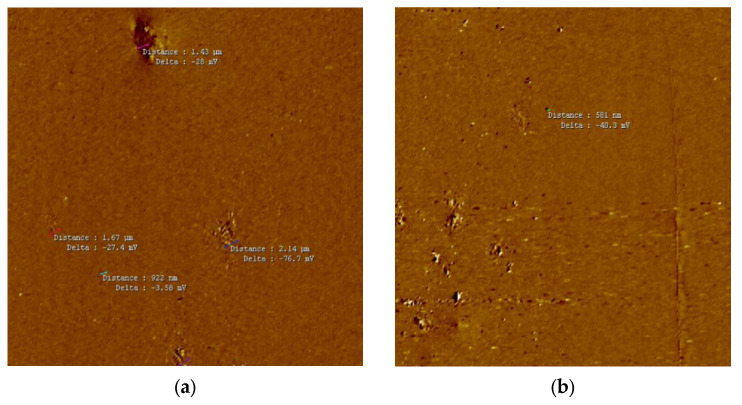
Atomic Force Microscopy (AFM) 2D amplitude of (**a**) Verdejo must control and (**b**) Verdejo must treated with UHPH.

**Figure 3 microorganisms-13-02623-f003:**
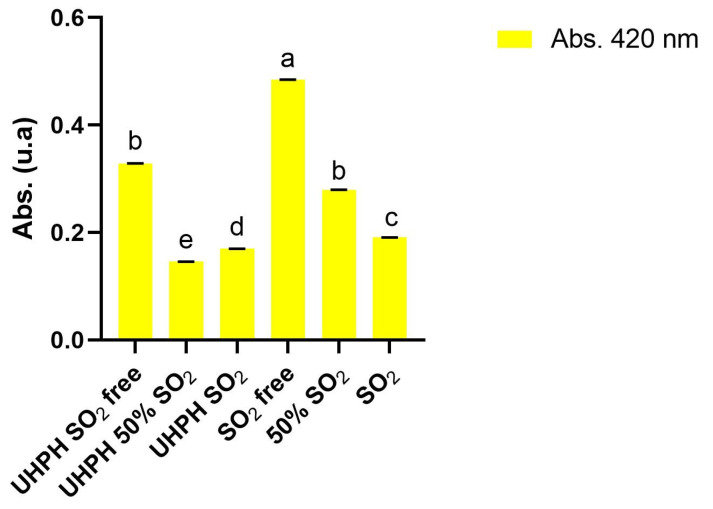
Absorbance at 420 nm of each must condition after clarification and UHPH treatment in the cases indicated. Data are expressed as the mean values (standard deviation) of n = 2 by ANOVA and Tukey’s HSD post-test (*p* < 0.05), and statistically significant differences are indicated by different letters (a, b, c, d, e). The nomenclature “free” indicates that no sulphur dioxide was added at the grape, must, or wine stages, the nomenclature “SO_2_ 50%” indicates that grape and must were treated with 3 g/hL of sodium metabisulphite, and “SO_2_” indicates that grape and must were treated with 6 g/hL of sodium metabisulphite. UHPH indicates the sets treated by UHPH.

**Figure 4 microorganisms-13-02623-f004:**
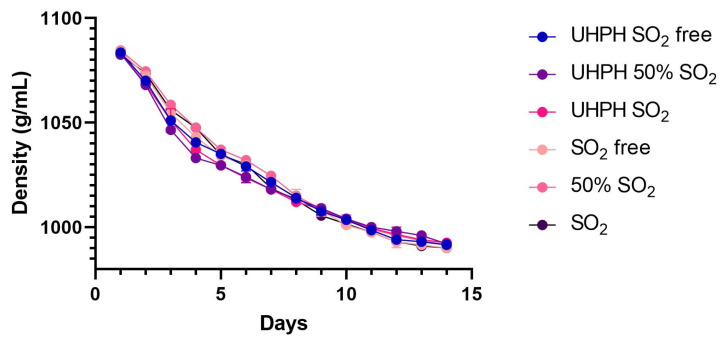
Density evolution for each studied alcoholic fermentation. Mean data are shown. The nomenclature “free” indicates that no sulphur dioxide was added at the grape, must, or wine stages, the nomenclature “SO_2_ 50%” indicates that grape and must were treated with 3 g/hL of sodium metabisulphite, and “SO_2_” indicates that grape and must were treated with 6 g/hL of sodium metabisulphite. UHPH indicates the sets treated by UHPH.

**Figure 5 microorganisms-13-02623-f005:**
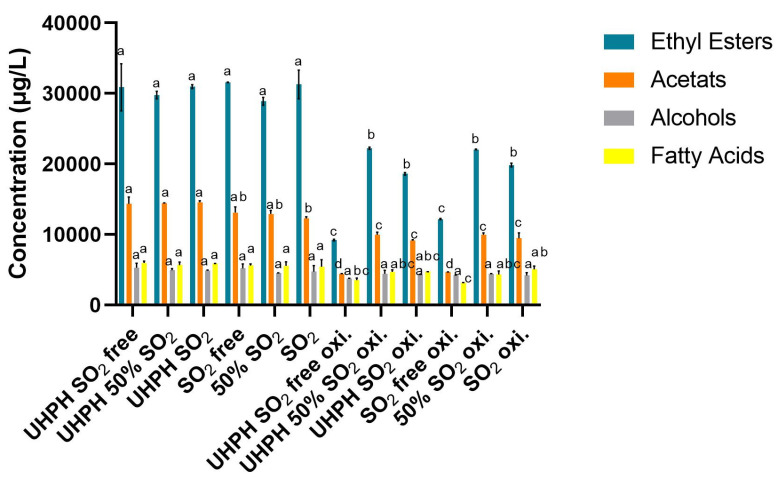
Concentrations of total fermentative aromas (ethyl esters: ethyl butyrate, ethyl isovaleriate, ethyl hexanoate, ethyl octanoate, ethyl decanoate, ethyl dodecanoate, diethyl succinate, and ethyl acetate; acetates: isobutyl acetate, hexyl acetate, and 2-phenylethyl acetate; alcohols: isobutanol, isoamyl alcohol, benzil alcohol, and 2-phenylethyl alcohol; and fatty acids: hexanoic acid, octanoic acid, and decanoic acid) for each family after alcoholic fermentation for all the studied treatments and after the accelerated aging process. Data are expressed as mean values (standard deviation) of n = 2 by ANOVA and Tukey’s HSD post-test (*p* < 0.05), and statistically significant differences are indicated by different letters (a, b, c, and d) for each different fermentative aromas families. “oxi” refers to wines submitted to forced oxidation. All results are expressed as the equivalent of 2-octanol in μg/L. Supplementation data is shown in Table A1 in Appendix A.

**Figure 6 microorganisms-13-02623-f006:**
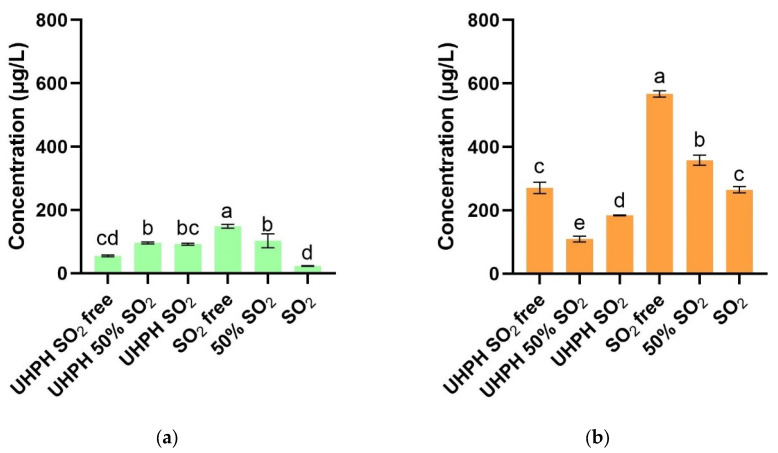
(**a**) Total oxidative aromas in Verdejo’s wines before the oxidation process and (**b**) total oxidative aromas in Verdejo’s wines after the oxidation process. Data are expressed as mean values (standard deviation) of *n* = 2 by ANOVA and Tukey’s HSD post-test (*p* < 0.05), and significant differences are indicated by different letters (a, b, c, d and e).

**Figure 7 microorganisms-13-02623-f007:**
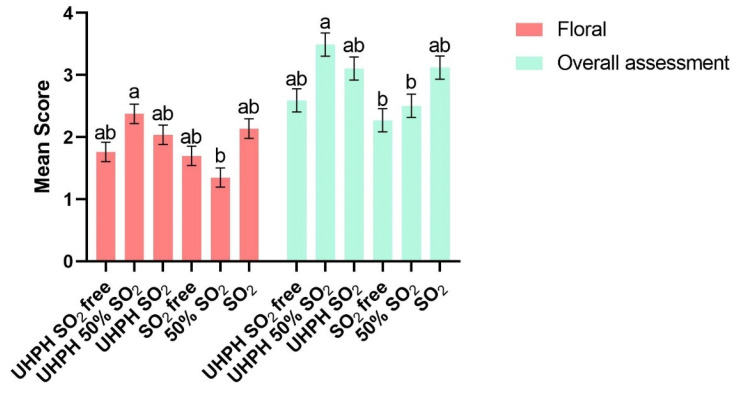
Mean scores and significant differences obtained in the floral and overall assessment of wines performed by tasting panels. Data are expressed as mean values (standard deviation) of *n* = 2 by ANOVA and Tukey’s HSD post-test (*p* < 0.05), and statistically significant differences are indicated by different letters applied to bars of the same color (a and b).

**Figure 8 microorganisms-13-02623-f008:**
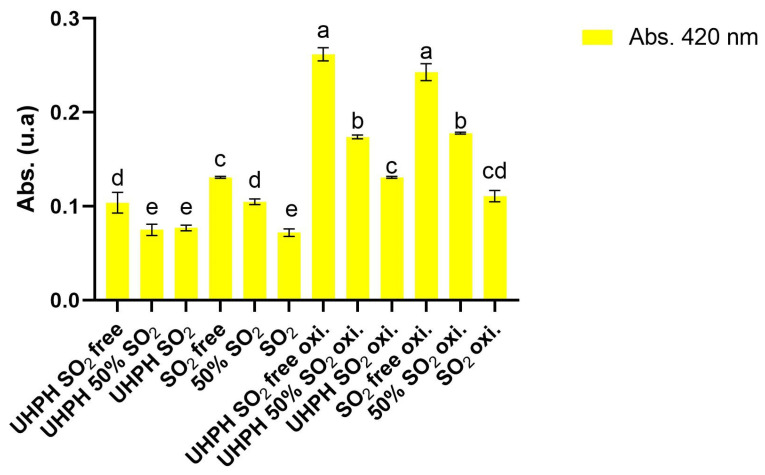
Absorbance at 420 nm of each wine before and after the oxidation process. Data are expressed as mean values (standard deviation) of *n* = 2 by ANOVA and Tukey’s HSD post-test (*p* < 0.05), and significant differences are indicated by different letters (a, b, c, d and e).

**Table 1 microorganisms-13-02623-t001:** AFM sizes of the colloidal particles in the UHPH-processed and control Verdejo juices. Data are expressed as the mean values and SD (standard deviation) of *n* = 2.

	Verdejo		
	2023	2024	2024
Measure	UHPH (nm)	UHPH (nm)	Control (nm)
Mean ± SD	518.6 ± 129.3	494.8 ± 143.7	2016.0 ± 781.0
Min.	312.0	183.2	1127.0
Max.	973.9	843.6	3759.0

**Table 2 microorganisms-13-02623-t002:** Basic chemical characterization of Verdejo must after clarification and UHPH batch treatment. Data are expressed as the mean values (standard deviation) of *n* = 2 by ANOVA and Tukey’s HSD post-test (*p* < 0.05), and statistically significant differences are indicated by different letters in the same row (a, b, c, d, e, f).

	UHPH SO_2_-Free	UHPH 50% SO_2_	UHPH SO_2_	SO_2_-Free	50% SO_2_	SO_2_
°Brix	22.7 a	22.8 a	22.7 a	22.7 a	22.8 a	22.7 a
Glucose + Fructose (g/L)	222.8 a	223.9 a	228.8 a	222.8 a	223.9 a	222.8 a
Potential Alcohol Strength (% vol.)	13.24 a	13.31 a	13.24 a	13.24 a	13.31 a	13.44 a
Total Acidity (g/L)	4.61 a	4.30 a	4.42 a	4.69 a	4.46 a	4.48 a
pH	3.46 a	3.47 a	3.45 a	3.44 a	3.43 a	3.42 a
Ammonia (mg/L)	110 a	113 a	113.0 a	113 a	114 a	100 a
Primary Amino Nitrogen (mg/L)	50 a	53 a	52 a	51 a	48 a	37 a
Yeast-Assimilable Nitrogen (mg/L)	149 a	154 a	154 a	153 a	151 a	129 a
L-Malic acid (g/L)	1.44 a	1.44 a	1.40 a	1.39 a	1.43 a	1.48 a
Abs. 420 nm	0.329 b	0.146 f	0.170 e	0.485 a	0.280 c	0.191 d

**Table 3 microorganisms-13-02623-t003:** Oenological parameters of wines from must processed or untreated by UHPH. Data is expressed as the mean values (standard deviation) of *n* = 2 by ANOVA and Tukey’s HSD post-test (*p* < 0.05), and significative differences are indicated by different letters in the same row (a, b, c, and d).

	UHPH SO_2_-Free	UHPH 50% SO_2_	UHPH SO_2_	SO_2_-Free	50% SO_2_	SO_2_
Alcohol (% vol.)	14.15 a	14.36 a	14.26 a	14.35 a	14.43 a	14.43 a
Volatile acidity (g/L)	0.45 a	0.48 a	0.41 a	0.45 a	0.48 a	0.47 a
Total acidity (g/L)	4.4 a	4.4 a	4.4 a	4.3 a	4.3 a	4.4 a
pH	3.40 a	3.41 a	3.41 a	3.43 a	3.43 a	3.42 a
L-Malic acid (g/L)	1.1 a	1.1 a	1.1 a	1.1 a	1.1 a	1.1 a
L-Lactic acid (g/L)	<0.1 a	<0.1 a	<0.1 a	<0.1 a	<0.1 a	<0.1 a
Abs. 420 nm	0.104 b	0.075 c	0.077 c	0.072 c	0.105 b	0.131 a
L*	98.85 a	99.65 a	99.15 a	99.65 a	98.05 a	97.75 a
a*	−1.36 a	−1.41 a	−1.20 a	−1.35 a	−1.00 a	−1.12 a
b*	7.95 ab	6.40 cd	5.84 d	5.68 d	7.32 bc	8.76 a
Turbidity	1.03 c	3.89 bc	1.73 c	7.41 ab	11.8 a	9.53 a
Folin–Ciocalteu index	3.85 a	4.15 a	4.33 a	6.11 a	5.08 a	3.91 a

**Table 4 microorganisms-13-02623-t004:** Microbiological results by culture plate and RT-PCR for all wines after bottling for all treatments. Data are expressed as mean values and SD (standard deviation) of *n* = 2. Data is expressed as the mean values (standard deviation) of *n* = 2 by ANOVA and Tukey’s HSD post-test (*p* < 0.05), and significative differences are indicated by different letters in the same row (a and b).

	UHPH SO_2_-Free	UHPH 50% SO_2_	UHPH SO_2_	SO_2_-Free	50% SO_2_	SO_2_
Yeast (cfu/100 mL)	<1 b	<1 b	<1 b	<1 b	240 a	<1 b
Lactic bacteria (cfu/100 mL)	<1	<1	<1	<1	<1	<1
Acetic acid bacteria (cfu/100 mL)	<1 b	<1 b	<1 b	<1 b	18 a	<1 b
*Brettanomyces* (cfu/100 mL)	<1 b	<1	< 1	<1	<1	<1
Yeast (cells/mL)	118 b	103 b	105 b	109 b	20.825 a	119 b
Lactic acid bacteria (cells/mL)	<20	<20	<20	<20	<20	<20
Acetic acid bacteria (cells/mL)	<20	<20	<20	<20	56	<20
*Brettanomyces bruxellensis* (cells/mL)	<20	<20	<20	<20	<20	<20

**Table 5 microorganisms-13-02623-t005:** Concentration of oxidation markers by families for all studied wines (UHPH SO_2_-free; UHPH 50% SO_2_; UHPH SO_2_; SO_2_-free; 50% SO_2_; and SO_2_) after bottling “Initial” and after an accelerated aging process “Oxidated”. All data are expressed in µg/L. Data are expressed as the mean values of *n* = 2 by ANOVA and Tukey’s HSD post-test (*p* < 0.05), and significant differences are indicated by different letters in the same row (a, b, c, d, e and f). n.d.: not detected.

	UHPH SO_2_-Free		UHPH 50%SO_2_		UHPH SO_2_		SO_2_-Free		50% SO_2_		SO_2_	
	Initial	Oxidated	Initial	Oxidated	Initial	Oxidated	Initial	Oxidated	Initial	Oxidated	Initial	Oxidated
(E)2-hexenal	0.5 f	3.0 d	1.7 e	4.3 c	1.6 ef	4.0 cd	8.1 a	6.1 b	0.9 ef	4.4 c	0.9 ef	7.4 a
(E)2-heptenal	0.1 e	0.8 b	0.5 c	0.3 cd	0.5 c	0.3 cd	0.4 cd	1.2 a	0.3 d	0.8 b	0.9 b	0.8 b
(E)2-octenal	9.2 c	4.6 d	10.7 c	2.7 d	2.3 d	0.5 d	20.4 a	0.6 d	13.0 bc	2.2 d	1.2 d	15.7 b
(E)2-nonenal	0.9 cd	1.1 cd	1.6 c	3.6 a	0.3 d	0.8 cd	2.8 b	2.0 c	3.3 a	3.3 a	3.1 b	1.2 cd
Total alkenals	10.7	9.6	14.4	10.9	4.7	5.7	31.6	9.9	17.5	10.6	6.1	25.0
3-methylbutanal	n.d.	129.0 b	n.d.	14.3 e	n.d.	89.3 c	n.d.	294.2 a	n.d.	71.5 d	n.d.	16.4 e
2-phenylacetaldehyde	15.9 f	72.8 b	36.2 de	46.3 d	17.3 f	43.3 d	28.3 e	126.0 a	11.1 f	36.3 de	12.1 f	60.7 c
methional	n.d.	n.d.	n.d.	n.d.	n.d.	n.d.	n.d.	n.d.	n.d.	n.d.	n.d.	n.d.
Total strecker aldehides	15.9	191.3	36.2	60.6	17.3	132.6	28.3	420.2	11.1	107.8	12.1	77.0
γ-nonalactona	13.8 e	14.2 e	20.3 bcd	17.2 cde	13.3 e	15.8 e	20.6 bcd	21.5 bc	16.6 de	57.0 a	1.6 f	22.9 b
Total lactones	13.8	14.2	20.3	17.2	13.3	15.8	20.6	21.5	16.6	57.0	1.6	22.9
3-methyl-2,4-nonadiona	0.6 e	4.7 de	11.4 c	13.9 c	0.5 e	18.4 c	5.0 de	14.1 c	2.7 de	96.2 a	n.d.	72.3 b
Total ketones	0.6	4.7	11.4	13.9	0.5	18.4	5.0	14.1	2.7	96.2	n.d.	72.3
4,5-Dimethyl-3-hydroxy-2,5-dihydrofuran-2-one	n.d.	n.d.	n.d.	n.d.	n.d.	n.d.	n.d.	n.d.	n.d.	n.d.	n.d.	n.d.
Total furans	n.d.	n.d.	n.d.	n.d.	n.d.	n.d.	n.d.	n.d.	n.d.	n.d.	n.d.	n.d.
Benzaldehyde	14.3 e	50.9 d	13.3 e	6.6 e	55.7 d	11.2 e	62.0 d	101.1 ab	54.9 d	86.0 bc	119.7 a	68.2 cd
Total aldehydes	14.3	50.9	13.3	6.6	55.7	11.2	62.0	101.1	54.9	86.0	119.7	68.2

**Table 6 microorganisms-13-02623-t006:** Color characteristics of wines processed or untreated by UHPH after forced oxidation processes. Data are expressed as mean values (standard deviation) of n = 2 by ANOVA and Tukey’s HSD post-test (*p* < 0.05), and significant differences are indicated by different letters (a, b, c, and d).

	UHPH SO_2_-Free	UHPH 50% SO_2_	UHPH SO_2_	SO_2_-Free	50% SO_2_	SO_2_
Abs. 420 nm	0.262 a	0.174 b	0.131 c	0.243 a	0.178 b	0.111 c
L*	94.55 d	97.35 ab	98.25 a	95.40 cd	96.60 bc	98.30 a
a*	−0.80 a	−0.84 a	−0.91 a	−1.03 a	−0.66 a	−0.59 a
b*	15.39 a	11.56 b	8.82 c	14.38 a	10.95 b	7.15 c
Folin–Ciocalteu index	2.84 b	3.16 b	3.38 b	3.34 b	3.46 b	4.07 a

**Table 7 microorganisms-13-02623-t007:** Fermentative aroma concentrations for the studied treatments before and after oxidation processes. Data are expressed as mean values (standard deviation) of *n* = 2 by ANOVA and Tukey’s HSD post-test (*p* < 0.05), and significant differences are indicated by different letters in the same row (a, b, c, and d). All results are expressed as equivalents of 2-octanol in μg/L. Supplementation data is shown in Table A1 in Appendix A.

	Sample	Total Ethyl Esters	Total Acetates	Total Alcohols	Total Fatty Acids
Initial sample	UHPH SO_2_-free	30.874 a	14.336 a	5.275 a	5.958 a
	UHPH 50% SO_2_	29.779 a	14.448 a	4.912 a	5.634 a
	UHPH SO_2_	30.977 a	14.598 a	4.835 a	5.872 a
	SO_2_-free	31.594 a	13.124 ab	5.189 a	5.581 a
	50% SO_2_	28.874 a	12.933 ab	4.519 a	5.541 a
	SO_2_	31.282 a	12.280 b	4.723 a	5.411 a
Forced oxidation process	UHPH SO_2_-free	9.198 c	4.372 d	3.766 a	3.530 bc
	UHPH 50% SO_2_	22.224 b	9.951 c	4.412 a	4.698 abc
	UHPH SO_2_	10.690 b	9.113 c	4.274 a	4.705 abc
	SO_2_-free	15.802 c	4.640 d	4.183 a	3.133 c
	50% SO_2_	22.060 b	9.966 c	4.413 a	4.316 abc
	SO_2_	15.433 b	9.457 c	4.174 a	5.040 ab

Fermentative aromas were formed by the sum of ethyl esters: ethyl butyrate, ethyl isovaleriate, ethyl hexanoate, ethyl octanoate, ethyl decanoate, ethyl dodecanoate, diethyl succinate, and ethyl acetate; acetates: isobutyl acetate, hexyl acetate, and 2-phenylethyl acetate; alcohols: isobutanol, isoamyl alcohol, benzil alcohol, and 2-phenylethyl alcohol; and fatty acids: hexanoic acid, octanoic acid, and decanoic acid.

## Data Availability

The original contributions presented in this study are included in the article. Further inquiries can be directed to the corresponding author.

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
