# Peer review of "Effect of UHPH and Sulphur Dioxide Content on Verdejo Vinification: Sensory, Chemical, and Microbiological Approach After Accelerated Aging Test"

_microorganisms, 2025, doi:10.3390/microorganisms13112623_

Round 1

Reviewer 1 Report

Comments and Suggestions for Authors

The manuscript provides novel information about the use of Ultra High-Pressure Homogenization in Verdejo must and the impact on chemical, microbiological and sensory characteristics of the resulting wine. The manuscript is interesting and of scientific value. The manuscript is clear and easy to understand, however, there are numerous spelling, grammatical, punctuation and formatting issues that need to be corrected. The authors must include some of the main findings in the abstract. The authors need to pay attention to their word selection and phrasing, which makes their sentences difficult to read and understand. There are parts of the results and discussion section where the authors did not explain how their results compare to literature. Figures and tables need minor corrections. Please see attachment for specific details.

Comments on the Quality of English Language

Language editing is recommended. The long sentence and some phrasing issues made the manuscript difficult to read and understand.

Author Response

Dear reviewer,

I would like to sincerely thank you for their valuable time, insightful comments, and constructive suggestions. Their detailed feedback has significantly contributed to improving the clarity, rigor, and overall quality of this manuscript. The reviewer’s observations have helped me to better structure my arguments, refine the interpretation of the results, and strengthen the discussion. I truly appreciate the effort and expertise that went into reviewing this work, which has had a positive impact on the development of present manuscript.

I have reviewed and modified each and every one of the corrections made. The most important ones made by you or the other reviewer are marked in blue in the text. Are detailed below some of the point comment on:

- Comment 1: According to the section 2.2 Chemical analysis. More details about methods and equiments used has been added in the direction to give more information to the reader,

- Comment 2: According to the sentence 535-537, it has been modified for a better explication, text marked in blue colour.

- Comment 3: According to the sentence 550-551, it has been modified for a better explication, text marked in blue colour.

- Comment 4: According to the sentence 709-711 in conlusions, it has been modified for a better explication, text marked in blue colour.

According to the comment about the "English could be improved to more cleraly express the research", this manuscript has been reviewed by an external professional to improve the overall clarity and quality of the written expression. In any case, we continue working to improve the second part of the article.

Attached you can find the revised manuscript with all the changes.

Once again, I sincerely thank you for their thoughtful and constructive feedback, which has greatly contributed to improving the quality and clarity of this work. I remain fully open to providing any further clarifications or additional information that might be helpful in supporting the review process.

Reviewer 2 Report

Comments and Suggestions for Authors

The proposed manuscript evaluates the use of UHPH on Verdejo winemaking, also in relation with increasing contents of sulfur dioxide used in conjunction or opposed (control) to pressure treatments. The topic is of interest for the wine field (reduction of sulfite contents in wines), the aims are well defined, the methodology appropriate but needs improvement in its description. In general, there are many aspects that require clarification or modifications, as indicated below.

- Lines 37-40: there is some confusion regarding the limits of sulfur dioxide: wrong limits are reported, and OIV limits are discussed but the EC regulation was referenced. Please correct.
- Lines 113-114: "metabisulphite" is not a complete compound name, please correct where ncessary.
- Section 2.2: multiple OIV methods are available for some analytes, therefore please list the OIV method names for each method you used (e.g. OIV-MA-AS313-15 for pH)
- Section 2.7: please indicate the volume of grape juice used in each independent fermentation.
- Volatile compound analysis (section 2.9; Figure 5, Table 7): no list of fermentative volatile compounds detected is available in the manuscript or in supplementary materials. Furthermore, the quantification of compounds was reported to be performed "by standards (R² > 0.99) and by internal standard (100 μL 2-octanol)" (what concentration of IS solution? on how much volume of wine?). The cited reference [32] cites in turn a reference by Torrens and co-workers (2004; https://doi.org/10.1093/chromsci/42.6.310) and therefore must be indicated the correct, original reference. Concerning detected compounds, please indicate the list of the compounds detected and which compounds were quantified with pure standard and which ones with a semi-quantitation on the basis of the internal standard.
- Sensory evaluation (section 2.10): please indicate how it was conducted the training phase. Please confirm that no informed consent and institutional review (e.g. ethical committee approval) were needed as indicated in lines 735-736.
- Section 3.1: 2023 samples were analyzed, but the study was carried out in 2024. Could you clarify the origin of these samples? (e.g. commercial samples, preliminary works)
- Lines 187, 308, 319: the threshold used to indicate the completeness of the fermentation is not coherent throughout the manuscript (0.5 vs 2.5 g/L), please correct.
- Sections 3.2 and 3.3 are placed in the wrong order and need to be inverted, otherwise the discussion firstly indicates the fermentation kinetics and then grape juice composition.
- Color intensity (or colour intensity, both listed in Table 2, 3, 6 and in the text): the evaluation of this parameter is not listed in Materials and Methods. Assuming the method is OIV-MA-AS2-07B [25], the authors are wrongly using this parameter because it can be applicable only to red and rosè wines (see section 2 of the cited method). Therefore, authors can keep the absorbance at 420 nm (reporting in the Methods how they analyzed it) and replace the color intensity in the text and tables (e.g. table 2, 3, 6) with this parameter. Furthermore, I would reconsider the use of absorbances at 520 and 620 nm in the study given that they are proper of red wines rather than white wines.
- Folin-Ciocateu determination (e.g. table 3, 6) was not included in the Materials and Methods
- The authors performed yeast and bacteria plate counts, highlighting the effects of the tested treatments. They noted some limited results (Table 4) compatible with the treatments performed, citing possible external contamination in some cases (lines 427-431, 446-447). However, it is unclear if all cellar and micro scale winemaking equipment was sanitized before treatments, and using what procedure. Please list the procedure performed for initial microbial sanitization in the Materials and Methods.
- Still on the topic of fermentative volatile compounds (Figure 5, Table 7), given that only aroma classes were used in the manuscript I think it would be necessary to make available the data for each compound and treatment as supplementary material (as done for oxidation markers in Table 5). It's data already available to authors and will be meaningful for readers to consult it.
- Table 7 presents few typos ("SO2", "Sampe") and does not present the units of measurement for the listed parameters.
- Few typos are present (e.g. line 566 "SO2" and "acceleration", line 546 "SO2", commas instead of points as in Table 6 fifth row second column)

Author Response

Dear reviewer,

I would like to sincerely thank you for their valuable time, insightful comments, and constructive suggestions. Their detailed feedback has significantly contributed to improving the clarity, rigor, and overall quality of this manuscript. The reviewer’s observations have helped me to better structure my arguments, refine the interpretation of the results, and strengthen the discussion. I truly appreciate the effort and expertise that went into reviewing this work, which has had a positive impact on the development of present manuscript.

I have reviewed and modified each one of the corrections made. All comments made by you or the other reviewer are marked in blue in the text. Are detailed below some of the point comment on:

  • Comment 1: Lines 37-40: there is some confusion regarding the limits of sulphur dioxide: wrong limits are reported, and OIV limits are discussed but the EC regulation was referenced. Please correct.
  • Response 1: Completely agree, there was a confusion. The limit fixed by OIV in red wine is 150.

  • Comment 2: Lines 113-114: "metabisulphite" is not a complete compound name, please correct where necessary.
  • Response 2: Completely agree, it is sodium metabisulphite.

  • Comment 3: Section 2.2: multiple OIV methods are available for some analytes, therefore please list the OIV method names for each method you used (e.g. OIV-MA-AS313-15 for pH)
  • Response 3: Completely agree and in the same direction that the other reviewer. I have added the detail of technique and equipment.
  •  
  • Comment 4: Section 2.7: please indicate the volume of grape juice used in each independent fermentation.
  • Response 4: done. 50L tanks for must clarification and 30 L for must fermentation.

  • Comment 5: Volatile compound analysis (section 2.9; Figure 5, Table 7): no list of fermentative volatile compounds detected is available in the manuscript or in supplementary materials. Furthermore, the quantification of compounds was reported to be performed "by standards (R² > 0.99) and by internal standard (100 μL 2-octanol)" (what concentration of IS solution? on how much volume of wine?). The cited reference [32] cites in turn a reference by Torrens and co-workers (2004; https://doi.org/10.1093/chromsci/42.6.310) and therefore must be indicated the correct, original reference. Concerning detected compounds, please indicate the list of the compounds detected and which compounds were quantified with pure standard and which ones with a semi-quantitation on the basis of the internal standard.
  • Response 5: done. The list of compounds was added in the tables and figures. The method for each compound identification was also detailed; it is as 2-octanol. And bibliography was changed by Torrens.

  • Comment 6: Sensory evaluation (section 2.10): please indicate how it was conducted the training phase. Please confirm that no informed consent and institutional review (e.g. ethical committee approval) were needed as indicated in lines 735-736.
  • Response 6: done. The judges of the tasting panel were selected, trained, and qualified following the normative ISO 8586:2012. The training consisted in familiarization with sensory evaluation procedures, trained to recognize and describe the basic tastes, aromas and textures and familiarization the use of scales and rating systems. Once formed all the judges must pass regular retraining and monitoring to ensure their skills remain sharp and results stable over time. All of them signed an informed consent form to participate in the sensory evaluation and no informed consent statement was required.

  • Comment 7: Section 3.1: 2023 samples were analysed, but the study was carried out in 2024. Could you clarify the origin of these samples? (e.g. commercial samples, preliminary works)
  • Response 7: 2023 results were preliminary works. The present study it is done with 2024 grapes.

  • Comment 8: Lines 187, 308, 319: the threshold used to indicate the completeness of the fermentation is not coherent throughout the manuscript (0.5 vs 2.5 g/L), please correct.
  • Response 8: done. In all the cases was 0,5 g/L.

  • Comment 9: Sections 3.2 and 3.3 are placed in the wrong order and need to be inverted, otherwise the discussion firstly indicates the fermentation kinetics and then grape juice composition.
  • Response 9: done. The order has been inverted.

  • Comment 10: Color intensity (or colour intensity, both listed in Table 2, 3, 6 and in the text): the evaluation of this parameter is not listed in Materials and Methods. Assuming the method is OIV-MA-AS2-07B [25], the authors are wrongly using this parameter because it can be applicable only to red and rosè wines (see section 2 of the cited method). Therefore, authors can keep the absorbance at 420 nm (reporting in the Methods how they analyzed it) and replace the color intensity in the text and tables (e.g. table 2, 3, 6) with this parameter. Furthermore, I would reconsider the use of absorbances at 520 and 620 nm in the study given that they are proper of red wines rather than white wines.
  • Response 10: done. We always use 420 520 and 620nm absorbances for colour characteristics for whites, res and rose. It is wll know, that for whites, 420nm it is the most important absorbance, at the end, the behaviour in 420nm and Color Intensity it is the same. We will follow this way in future studies.

  • Comment 11: Folin-Ciocateu determination (e.g. table 3, 6) was not included in the Materials and Methods.
  • Response 11: done. Total phenols were determined using the Folin-Ciocalteu assay (Method OIV-MA-AS2-10) with some modifications. Briefly, 100 μL of sample, 500 μL of Folin- Ciocalteu reagent and 2 mL of a sodiumcarbonate solution (1.88 M) were mixed, with final volume of 10 mL with water. The solution was stocked for 30 min for the reaction to take place and stabilize. Finally, the absorbance was measured at 750 nm by a Helios-α spectrophotometer (Thermo Fisher Scientific, Waltham, MA USA).

  • Comment 12: The authors performed yeast and bacteria plate counts, highlighting the effects of the tested treatments. They noted some limited results (Table 4) compatible with the treatments performed, citing possible external contamination in some cases (lines 427-431, 446-447). However, it is unclear if all cellar and micro scale winemaking equipment was sanitized before treatments, and using what procedure. Please list the procedure performed for initial microbial sanitization in the Materials and Methods.
  • Response 12: done. Experimental cellar facilities were cleaned and disinfected by hot water at 80ºC for 30 minutes and Ox-virin for all the surfaces (Grupo OX, Cuarte, Spain).

  • Comment 13: Still on the topic of fermentative volatile compounds (Figure 5, Table 7), given that only aroma classes were used in the manuscript I think it would be necessary to make available the data for each compound and treatment as supplementary material (as done for oxidation markers in Table 5). It's data already available to authors and will be meaningful for readers to consult it.
  • Response 13: done. We have all the data, so it is very easy for us to prepare it. We have added these information at as Appendix in the new manuscript.

  • Comment 14: Table 7 presents few typos ("SO2", "Sampe") and does not present the units of measurement for the listed parameters.
  • Response 14: done. Now all the SO2 mentions are in the same format and the units are detailed in the Table presentation.

  • Comment 15: Few typos are present (e.g. line 566 "SO2" and "acceleration", line 546 "SO2", commas instead of points as in Table 6 fifth row second column)
  • Response 15: done, now all data it is separed by points in place of commas.

According to the comment about the "English could be improved to more cleraly express the research", this manuscript has been reviewed by an external professional to improve the overall clarity and quality of the written expression. In any case, we continue working to improve the second part of the article.

Attached you can find the revised manuscript with all the changes.

Once again, I sincerely thank you for their thoughtful and constructive feedback, which has greatly contributed to improving the quality and clarity of this work. I remain fully open to providing any further clarifications or additional information that might be helpful in supporting the review process.

Round 2

Reviewer 1 Report

Comments and Suggestions for Authors

The manuscript has been improved but can still be improved. The authors should include some of the main results in the abstract. There is numerous spelling, grammatical, punctuation and formatting issues that need to be corrected. The figures and tables need minor corrections. Please see attachment for details.

Comments on the Quality of English Language

Language editing recommended.

Author Response

Dear reviewer,

I would like to sincerely thank you again for their valuable time and comments. All the comments increase the overall quality of this manuscript. I truly appreciate the effort reviewing this work.

I have reviewed and modified each one of the corrections proposed. All signifficant comments made are highlighted in blue in the text. The minor changes proposed are not indicated. More results have been added in the abstract. We have not had any comments from the reviewer 2.

The English has been carefully reviewed to imporve the quality of full manuscript removing all spelling, grammatical, punctuation mistakes.

According to the comment "Why it is sulphur dioxide written out in some cases ans SO2 used in other cases", the reason is: we used "sulphur dioxide" when we refer to the substance in general, describing its properties, used, effects, or others in descriptive or narrative text. In contrast, we used the chemical formula "SO2" in scientific or technical context or when we describe the treatments applied. We belive that this double nomenclature helps readers with the correct interpretation.

Please, do not hesitate to contact us for any other change or comment.

Best regards,

Miquel

Reviewer 2 Report

Comments and Suggestions for Authors

The authors replied to my comments and modified accordingly the manuscript.

Author Response

Thank you very much for your contributions in first revision, they have helped to make the proposed article better. I am glad to have resolved all your doubts. In any case, I am attaching the manuscript again with the latest changes proposed by reviewer 1. See you soon and do not hesitate to propouse any other change.

My best regards,

Miquel
